# Understanding the Mediating Role of Gastronomic Experiences in a World Heritage Site: An Explanatory Approach to the Case of Córdoba (Spain)

Laura Ortega-Pérez [1], Cándida María Domínguez-Valerio [2,*], Consolación Pascual-García [1] and María del Rosario Ruiz-Robles [1]

[1] Department of Statistics, Econometrics, Operational Research, Business Organization and Applied Economics, University of Córdoba, 14071 Córdoba, Spain; d92orpel@uco.es (L.O.-P.); z12pagac@uco.es (C.P.-G.); z42rurom@uco.es (M.d.R.R.-R.)

[2] Department of Economic and Social Science, Universidad Tecnológica de Santiago, Santiago de los Caballeros 51000, Dominican Republic

* Correspondence: candidadominguez1@docente.uesa.edu

## Abstract

Gastronomy is an essential element in the visitor's experience in a given destination, being even more important in those destinations with an important cultural heritage. The study focused on the city of Córdoba (Spain), known for its extensive cultural heritage and for being the city with the most World Heritage declarations in the world, with four of them. Based on a total of 467 valid questionnaires, a structural equation analysis (PLS-SEM) was carried out. Of a total of four hypotheses proposed, the influence of all of them was confirmed except for the one that hypothesized an influence of perceived gastronomic value on loyalty to a gastronomic destination, thus confirming the overall mediating effect of gastronomic experiences between the two variables mentioned above. Furthermore, the influence of gastronomic satisfaction on loyalty to a gastronomic destination was evident, with no mediating effect of gastronomic experiences in this case. The results obtained can be of great help to decision-makers, both public and private, in establishing gastronomic proposals that are genuine, sensorially rich, and strongly culturally rooted, thus enhancing the cultural experience.

**Keywords:** Córdoba; gastronomy; loyalty; cultural experience; gastronomic authenticity

## 1. Introduction

Currently, the culinary experience is a key factor in tourist satisfaction, especially in destinations with significant cultural heritage. This study analyses tourists' perceptions of gastronomy in Córdoba, a city with four UNESCO World Heritage Sites: three as World Heritage Sites (the Mosque, the Jewish Quarter, and Medina Azahara) and one as Intangible Heritage of Humanity (the Patio Festival). However, there are no empirical studies that analyse the mediating role of the gastronomic experience in the satisfaction of tourists arriving in the city of Córdoba, a city with important tourist activity and where nearly two million tourists arrive each year, which implies a strong economic impact in this place.

Gastronomy in a destination ranges from the basic need for food to a privileged way of learning about its history, customs, and social relationships. Consequently, understanding tourists' gastronomic experiences is essential to understanding their motivations and perceptions, as this can influence destination loyalty [1]. Indeed, gastronomy may serve

either as a complement to the travel experience or as a decisive factor in the selection of a destination [2,3], particularly within a context characterized by the pursuit of novel experiences [4]. The objective of this study is to test the mediating role of gastronomic experiences between perceived value/satisfaction and destination loyalty using SEM.

Various papers analyse the different fields of study in gastronomic tourism. Among others, we can highlight the studies by Anubha Hussein et al. (2023) [5], Kumar (2024) [6], Surjood et al. (2024) [7], and Park and Kim (2024) [8]. It is also necessary to relate three concepts: tourism, gastronomy, and culture [9]. Research in this field indicates two fundamental aspects [1,4,10]: first, tourists interested in gastronomy tend to have greater purchasing power; second, they are more demanding regarding the quality and authenticity of local food.

However, despite the growing importance of gastronomic tourism, studies analysing the gastronomic experience as a mediator of other variables, such as motivation or satisfaction, are limited. In this sense, the innovation of this research is to analyse the gastronomic experience as a mediator of satisfaction and loyalty to a destination.

## 2. Literature Review

### 2.1. Gastronomy as a Tourist Experience

Gastronomy, including wine [11], is an essential element of a visitor's experience at a given destination [12]. Therefore, it is necessary to analyse tourist behaviour within the destination itself and its relationship with the culinary heritage of that place [13]. However, not all tourists show interest in local cuisine. Thus, some visitors consider gastronomy only a basic need, while others consider it a way to learn about a place's culture and traditions and as part of their experience at the destination [14,15]. Thus, gastronomy can produce both positive and negative experiences for tourists in a given destination [16].

Tourist motivations for gastronomy can be classified into four categories [1]: first, physiological (basic nutrition); second, cultural (understanding the destination through its gastronomy); third, interpersonal (social interaction through food); and fourth, status and prestige (culinary experiences as a sign of distinction). These motivations can combine, leading some travellers to seek out different gastronomic experiences that determine their choice of a particular destination [3].

On the other hand, one line of research in gastronomic tourism is the analysis of the relationship between tourists and food at street stalls, especially in developing countries [17]. Various authors analyse this line of research. For example, Ghatak and Chattergee [18] analyse its food security, and Ukenna and Ayodele [19] consider it a key informal sector in some urban economies.

Gastronomic experiences are subjective and depend on multiple factors [20]. Quan and Wang [21] highlight that gastronomic experiences are based on the exploration of new ingredients and different ways of preparing and consuming food. Thus, taste and smell are determining factors in diner satisfaction [22], along with other factors such as the restaurant's ambiance and diners' interaction with restaurateurs [23].

The gastronomic experience is structured in three dimensions [23]: first, the appearance of the food (plating, colour, and texture); second, situational factors (restaurant ambiance and spatial layout); and third, individual factors (diner sensations and perceptions). Björk and Kauppinen-Räisänen [20] introduced the concept of "restaurantscape" to describe the interaction between gastronomy and the restaurant's ambiance, where elements such as atmosphere, flavour, and service quality play a crucial role. Likewise, it is necessary to strengthen gastronomic innovation in destinations [24].

Emotions are also key to the gastronomic experience. Desmet and Schifferstein [22] identify five influential variables: product appearance, product type, activity, context, and

memory. Consequently, gastronomy can be the primary reason for a trip or part of a broader cultural transmission. Thus, satisfaction is higher among travellers who enjoy a diverse gastronomic experience [14]. In this way, Muskat et al. (2024) [25] analyse the importance of both sensory and non-sensory factors in the tourist experience.

### 2.2. Gastronomy and Satisfaction

Growing competition among tourist destinations requires differentiation through authenticity and cultural aspects, as opposed to the standardization occurring in certain locations [26]. To achieve efficient and attractive management, a destination cannot base its offering solely on natural resources or widespread leisure activities. Today, tourists seek experiences that align with their expectations and demands, making gastronomy a key factor in satisfaction [27]. Tourist satisfaction with local gastronomy is essential, as it influences their loyalty to the destination, serving as a key element in the development of authentic and unforgettable experiences [28].

Hendijani [29] asserts that tourists' satisfaction with gastronomic experiences is profoundly influenced by the cultural depth of culinary traditions and the natural, healthful quality of ingredients, which are predominantly obtained from local producers. Furthermore, flavour plays a fundamental role in this experience [30], as many traditional recipes are part of the destination's cultural heritage and offer unique gastronomic experiences, distinct from those travellers find in their place of origin. Thus, gastronomy becomes a determining factor in tourist satisfaction [29].

Therefore, local gastronomy can be a key element in visitor satisfaction and destination perception [3,20]. This link between gastronomy, motivation, and the tourist experience reinforces the importance of developing authentic culinary proposals that enrich the tourist offerings and encourage traveller loyalty.

### 2.3. Perceived Value Through Local Gastronomy

The development of an attractive gastronomic offering in a destination can have a significant impact on tourism and other economic sectors. It also allows for the diversification of tourism activities and reduces seasonality in certain destinations [27]. To achieve this development, it is essential to implement an effective public-private policy that fosters innovative and sustainable culinary practices. This can be achieved through infrastructure improvements, the creation of gastronomic routes, and the promotion of differentiated culinary experiences [31]. Consequently, the integration of gastronomy with other activities, such as wine tourism or olive oil tourism, can enrich the visitor experience and make it unique and unforgettable [28].

However, for a gastronomic experience to be truly memorable, a destination's culinary offering must be both recognizable and identifiable. This implies the need to have an adequate number of gastronomic establishments that offer varied and authentic options, ensuring a satisfactory experience for tourists [32].

According to Jiménez Beltrán et al. [32], in their study on gastronomy in Córdoba (Spain), traditional cuisine is not only a key tourist attraction but also allows the destination's cultural heritage to be transmitted to visitors. Therefore, it is crucial to balance the preservation of culinary tradition with the incorporation of gastronomic innovations that enrich the offering without losing its essence [33]. Likewise, destinations need to develop strategies that reinforce the perceived value for tourists [34].

### 2.4. Gastronomy and Loyalty

Destination loyalty is a key aspect of marketing strategies, as it allows predicting consumer behaviour [35]. A loyal tourist represents a stable source of income and contributes to improving the destination's reputation through word of mouth (Baker and Crompton,

2000) [36]. Studies on tourist loyalty differentiate two main types: first, behavioural loyalty, which is reflected in repeat visits to the destination; and second, attitudinal or affective loyalty, which involves recommending the destination to others and the intention to return in the future [37]. Thus, loyalty can be defined as a tourist's commitment to a destination, manifested in their intention to return (behavioural loyalty) and their willingness to recommend it to others (attitudinal or affective loyalty). In this sense, previous gastronomic experiences are fundamental to tourists' loyalty to the destination [38]. Rousta and Jamshidi (2020) [38] point out that destination loyalty is closely related to the gastronomic experience, especially through taste/quality value, health value, price value, emotional value and prestige value.

According to the scientific literature, the hypotheses to be tested are as follows:

**H1a.** *Perceived gastronomic value influences loyalty to a gastronomic destination.*

**H1b.** *The relationship between perceived gastronomic value and loyalty to a gastronomic destination is positively mediated by gastronomic experiences at the destination.*

**H2a.** *Gastronomic satisfaction influences loyalty to a gastronomic destination.*

**H2b.** *The relationship between gastronomic satisfaction and loyalty to a gastronomic destination is positively mediated by gastronomic experiences at the destination.*

Figure 1 presents the proposed theoretical structural model.

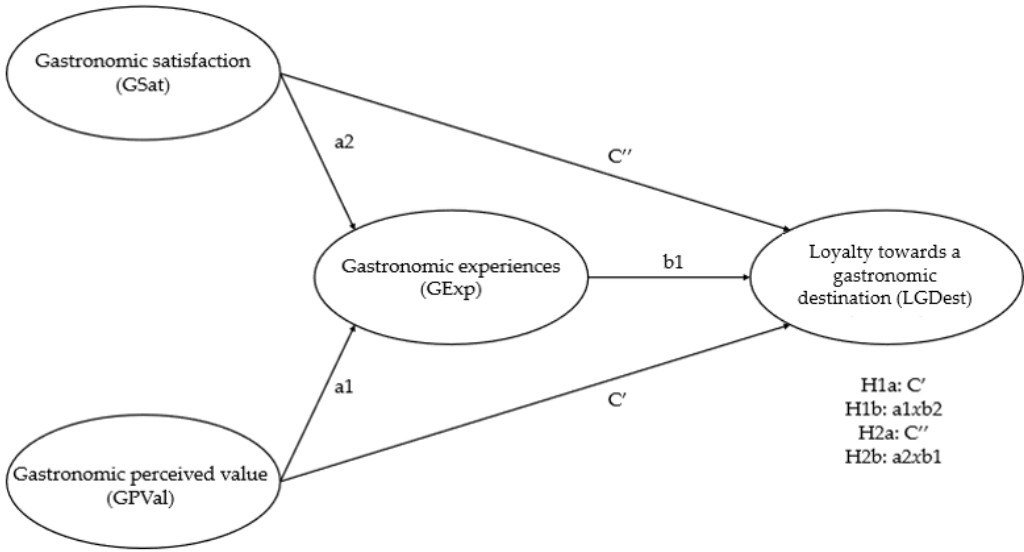

**Figure 1.** Theoretical structural model.

## 3. Methodology

### 3.1. Questionnaire Design

To achieve the objectives set forth in this research, a quantitative methodology was adopted based on the application of a questionnaire structured in three sections. The first section analyses the respondents' motivations for their trips. The second addresses aspects related to the perceived value of gastronomy, gastronomic experiences, culinary satisfaction, and loyalty to a gastronomic destination. The third section collects information on the participants' sociodemographic profile. The questions in the first section were formulated in a polychromous format, while the second section used 5-point Likert scales, where 1 represented "strongly disagree" and 5 "strongly agree". The questions in the

sociodemographic section were structured in dichotomous (e.g., gender), polychromous (e.g., educational level, professional category, or income level), and open-ended formats (e.g., age).

The survey utilised in this research was grounded in previous studies and research. Thus, the satisfaction items are found in the previous literature of Cracolici et al. (2008) [39], Haven-Tang and Jones (2005) [28], and Hendijani (2016) [29]. Meanwhile, the gastronomic experience items are found in the previous literature of Björk and Kaup-pinen-Räisänen (2017) [20] and Taar (2014) [23]. Meanwhile, the perceived value of gastronomy items come from previous studies by Ignatov and Simith (2006) [31] and Jiménez Beltrán et al. (2016) [32]. The loyalty and gastronomy items are derived from the study by Chen and Chen (2009) [35]. Finally, the tourist sociodemographic profile items are found in the contributions of Park (2017) [10] and Robinson et al. (2018) [33].

The initial questionnaire validation process went through several stages (Moore et al., 2021) [40]. Initially, the proposed items were analysed by a panel of professional experts in tourism and gastronomy. The questionnaire was then sent to various tourism researchers. The final stage of this validation consisted of conducting a pilot study with 25 people. These three phases allowed for verification of both the clarity of the items and the suitability of the fieldwork process. The objective was to design a questionnaire that was easily understandable for respondents and that did not generate any questions (Hair et al., 2022) [41]. This approach is essential for obtaining reliable and high-quality results, minimizing the risk of problems during data collection during fieldwork [40].

*3.2. Fieldwork*

Fieldwork was conducted between June 2022 and June 2023 with tourists visiting the city of Córdoba (Spain), yielding a total of 481 questionnaires. The questionnaire was provided in both English and Spanish to reach the largest possible number of respondents. Convenience sampling was used. After a filtering process in which those with more than 15% missing data were eliminated [41], the final sample size was 467 questionnaires. To validate the adequacy of the sample size, the G*Power 3 program [42] was used, determining a required minimum of 348 questionnaires, a figure that was significantly exceeded in this research. The results justifying this sample size are presented in Table 1.

**Table 1.** Relevant sample size.

| F Tests—Linear Multiple Regression: Fixed Model, $R^2$ Deviation from Zero | | |
|---|---|---|
| Analysis: | A Priori: Compute Required Sample Size | |
| Input: | Effect size $f^2$ | =0.05 |
| | $\alpha$ err prob | =0.05 |
| | Power (1 − β err prob) | =0.95 |
| | Number of predictors | =3 |
| Output | Noncentrality parameter λ | =17.4000000 |
| | Critical F | =2.6308673 |
| | Numerator df | =3 |
| | Denominator df | =344 |
| | Total sample size | =348 |
| | Actual power | =0.9503964 |

The questionnaire was distributed at the most important tourist spots in the city of Córdoba. Before the questionnaire was completed, the survey team asked a preliminary question (Have you tried the local cuisine of Córdoba?). If the response was positive, the questionnaire was continued; if not, it was not.

### 3.3. Bias-Control Methods

A critical aspect to be addressed in empirical research pertains to methodological validation, which can be undermined by biases in participants' responses. In order to minimize such biases, the analysis was conducted employing a pre-post methodology. In the initial phase, procedural controls were implemented, including the adoption of familiar and unambiguous language, the use of straightforward vocabulary, and the deliberate avoidance of complex syntactic structures that might otherwise lead to misinterpretation of the measurement indicators. Moreover, participants were explicitly informed that their responses would remain entirely anonymous and that there were no objectively correct or incorrect answers [43,44].

In the post-test phase, Harman's single-factor test [45] was applied to evaluate the potential presence of common method bias. According to this test, if a single factor accounts for more than 50% of the total variance, it suggests possible bias issues. As shown in Table 2, the results indicate that such concerns are not present, given that the percentage of variance explained by the single factor is 40.607%, which falls below the critical threshold.

**Table 2.** Harman test.

| Total Variance Explained | | | | | | |
|---|---|---|---|---|---|---|
| | | Initial Eigenvalues | | | Extraction Sums of Squared Charges | |
| Component | Total | % Variance | % Accumulated | Total | % Variance | % Accumulated |
| 1 | 16.649 | 40.607 | 40.607 | 16.649 | 40.607 | 40.607 |

### 3.4. Preliminary Analysis of the Data and Sociodemographic Profile of the Sample

The preliminary data analysis involved the calculation of the mean, standard deviation, the Kolmogorov–Smirnov test for normality, and Cronbach's alpha to assess the reliability of the measurement scale. The corresponding results are presented in Table 3. Notably, the findings indicate that the data do not follow a normal distribution, thereby classifying them as non-parametric. Concerning the reliability of the scale, the results confirm an optimal level of internal consistency, as all construct-specific values exceed the threshold of 0.7 [46].

**Table 3.** Preliminary analysis of the data.

| | Media | D.T. | Test K-S | Cronbach |
|---|---|---|---|---|
| Gastronomic perceived value (GPVal) | | | | 0.862 |
| *GPVal1—Quality of the dishes* | 4.31 | 0.852 | 0.000 [C] | |
| *GPVal2—Variety of dishes* | 4.08 | 0.933 | 0.000 [C] | |
| *GPVal3—Prices* | 4.11 | 0.963 | 0.000 [C] | |
| *GPVal4—Culinary establishment facilities* | 3.73 | 0.976 | 0.000 [C] | |
| *GPVal5—Establishment atmosphere* | 4.08 | 0.936 | 0.000 [C] | |
| *GPVal6—Innovation and new flavours in dishes* | 3.65 | 1.137 | 0.000 [C] | |
| *GPVal7—Service and hospitality* | 4.52 | 0.791 | 0.000 [C] | |
| *GPVal8—Traditional cuisine* | 4.29 | 0.869 | 0.000 [C] | |
| Gastronomic experiences (GExp) | | | | 0.900 |
| *GExp1—Authentic culinary experience* | 3.91 | 0.955 | 0.000 [C] | |
| *GExp2—Understanding the culture and traditions of the destination* | 4.23 | 0.843 | 0.000 [C] | |
| *GExp3—Good smell* | 4.14 | 0.925 | 0.000 [C] | |
| *GExp4—Good visual appearance* | 4.17 | 0.897 | 0.000 [C] | |
| *GExp5—Good taste* | 4.46 | 0.865 | 0.000 [C] | |
| *GExp6—Fresh ingredients* | 4.37 | 0.894 | 0.000 [C] | |
| *GExp7—Different flavours from the food where I live* | 4.15 | 0.945 | 0.000 [C] | |
| *GExp8—Different from what I normally eat* | 4.15 | 0.964 | 0.000 [C] | |

**Table 3.** *Cont*.

| | Media | D.T. | Test K-S | Cronbach |
|---|---|---|---|---|
| Gastronomic satisfaction (GSat) | | | | 0.847 |
| *GSat1—Gastronomy is important for my level of satisfaction with a destination* | 4.01 | 1.002 | 0.000 [C] | |
| *GSat2—It was a good choice to taste the gastronomy at this destination* | 4.32 | 0.850 | 0.000 [C] | |
| *GSat3—My level of satisfaction with the gastronomy at this destination was important* | 4.19 | 0.947 | 0.000 [C] | |
| Loyalty towards a gastronomic destination (LGDest) | | | | 0.861 |
| *LGDest1—I would recommend a visit to a destination if someone asked me for advice regarding its cuisine* | 4.29 | 0.907 | 0.000 [C] | |
| *LGDest2—I will encourage my family and friends to visit certain restaurants* | 4.22 | 0.948 | 0.000 [C] | |
| *LGDest3—When I like a restaurant, I try to return to that destination* | 4.09 | 1.060 | 0.000 [C] | |
| *LGDest4—I intend to buy local food products that I tried during that trip* | 3.86 | 1.124 | 0.000 [C] | |
| *LGDest5—I will recommend local food products from the destinations I visit* | 4.19 | 1.022 | 0.000 [C] | |

Notes: C: Lilliefors significance correction; SD: standard deviation; K-S test: Kolmogorov–Smirnov normality test.

Regarding the sociodemographic profile, this corresponds to a Spanish woman, under 25 years old, with university studies, who is either a student or a public sector employee, with an income level between EUR 1501 and EUR 2500. The sociodemographic profile shows a predominance of young people interested in gastronomy.

Table 4 presents the more detailed results of the sociodemographic profile.

**Table 4.** Sociodemographic profile.

| Variable | % | Variable | % | Variable | % |
|---|---|---|---|---|---|
| Gender | | Level of education | | Professional category | |
| Man | 40.1 | Primary education | 1.1 | Independent professional | 7.3 |
| Woman | 59.9 | Secondary education | 25.8 | Public employee | 23.7 |
| | | University degree | 37.5 | Private company worker | 20.4 |
| | | Postgraduate/master/PhD | 35.6 | Self-employed | 6.4 |
| | | | | Student | 37.4 |
| | | | | Unemployed | 3.9 |
| | | | | Retired | 0.3 |
| | | | | Housework | 0.6 |
| Age | | Net monthly income level | | Country of origin | |
| 18–25 years | 34.5 | Under 700 € | 11.8 | Spain | 43.4 |
| 25–35 years | 26.0 | 700 €–1000 € | 9.5 | Peru | 17.6 |
| 36–45 years | 18.3 | 1001 €–1500 € | 15.2 | France | 14.0 |
| 46–55 years | 15 | 1501 €–2500 € | 35.2 | United Kingdom | 7.46 |
| More than 55 years | 6.2 | 2501 €–3500 € | 10.3 | Others | 17.54 |
| | | More than 3500 € | 18.0 | | |

Note: Currency standardization: 1€ = 1.07$.

## 4. Results

Given the explanatory nature of the research [47], the focus of this study has been on the explanatory power and effect size of the endogenous variables, as well as on the statistical inference of structural relationships or testing of previously proposed hypotheses supported by previous literature. Gefen [48] identifies two distinct stages of analysis within the framework of structural equation modelling. The first stage involves examining and validating the measurement model by evaluating the relationships between indicators and their corresponding constructs. The second stage pertains to the structural model, wherein the interrelationships among the constructs themselves are analysed.

### 4.1. Analysis of the Measurement Model

The analysis of the measurement model at the individual level examines the factor loadings of the Mode A composite indicators that comprise the model. Thus, authors such as Ali et al. [49] establish a minimum level of 0.707 for factor loadings. However, other authors such as Barclay et al. [50] establish a more flexible approach and lower the minimum factor loading to 0.60, as long as maintaining this indicator does not imply a deterioration in content validity. In any case, a loading below 0.40 must be eliminated [51]. For the analysis of factor loadings, the associated significance values for these factor loadings have also been reported. In the present model, the GPVal3 indicator was eliminated because it had a factor loading below 0.60, and its elimination substantially improved the model's internal consistency.

At the internal consistency level, the rigour with which the indicators of a given construct are measuring the same latent variable or construct is tested. To this end, leading authors such as Dijkstra and Henseler [52] point out that the most reliable measure for testing such consistency is the Dijkstra–Henseler Rho_A. On the other hand, convergent validity is analysed through the average variance extracted (AVE), which must present AVE values equal to or greater than 0.50, which would imply that each construct contributes to explaining at least 50% of the variance of the assigned indicators. Authors such as Henseler et al. [53] establish a minimum Rho_A level of 0.70 to demonstrate acceptable internal consistency. In contrast, Hair et al. [54] propose the application of 95% confidence intervals as a means to evaluate the statistical significance of the coefficients pertaining to internal consistency, namely Rho_A and AVE in the present case. The results of the reliability and validity assessments of the measurement model are detailed in Table 5.

**Table 5.** Reliability and validity analysis.

|  | | Loads (Sig.) | Rho_A | CI95% | AVE | CI95% |
|---|---|---|---|---|---|---|
| Gastronomic perceived value (GPVal) | | | 0.854 | [0.819; 0.882] | 0.530 | [0.479; 0.583] |
| | GPVal1 | 0.773 (0.000) | | | | |
| | GPVal2 | 0.784 (0.000) | | | | |
| | GPVal4 | 0.758 (0.000) | | | | |
| | GPVal5 | 0.766 (0.000) | | | | |
| | GPVal6 | 0.654 (0.000) | | | | |
| | GPVal7 | 0.697 (0.000) | | | | |
| | GPVal8 | 0.651 (0.000) | | | | |
| Gastronomic experiences (GExp) | | | 0.903 | [0.873; 0.923] | 0.590 | [0.527; 0.647] |
| | Gexp1 | 0.729 (0.000) | | | | |
| | Gexp2 | 0.736 (0.000) | | | | |
| | Gexp3 | 0.835 (0.000) | | | | |
| | Gexp4 | 0.801 (0.000) | | | | |
| | Gexp5 | 0.840 (0.000) | | | | |
| | Gexp6 | 0.805 (0.000) | | | | |
| | Gexp7 | 0.688 (0.000) | | | | |
| | Gexp8 | 0.691 (0.000) | | | | |
| Gastronomic satisfaction (GSat) | | | 0.853 | [0.812; 0.882] | 0.768 | [0.723; 0.808] |
| | Gsat1 | 0.835 (0.000) | | | | |
| | Gsat2 | 0.893 (0.000) | | | | |
| | Gsat3 | 0.900 (0.000) | | | | |
| Loyalty towards a gastronomic destination (LGDest) | | | 0.889 | [0.863; 0.908] | 0.653 | [0.605; 0.699] |
| | LGDest1 | 0.849 (0.000) | | | | |
| | LGDest2 | 0.884 (0.000) | | | | |
| | LGDest3 | 0.703 (0.000) | | | | |
| | LGDest3 | 0.722 (0.000) | | | | |
| | LGDest5 | 0.865 (0.000) | | | | |

With respect to discriminant validity, this criterion assesses the degree to which a particular construct is empirically distinct from the other constructs comprising the structural model. Henseler et al. [53] asserted that the most effective measure for detecting the absence of discriminant validity is the heterotrait–monotrait ratio (hereafter referred to as HTMT). HTMT values exceeding 0.90 are indicative of insufficient discriminant validity [55]. In this research, and as with the Rho_A and the AVE, the associated 95% confidence intervals have been reported to test the significance of the HTMT values. The results are shown in Table 6. These results demonstrate the existence of discriminant validity between the different constructs that make up the structural model.

**Table 6.** Discriminant validity. Heterotrait–monotrait ratio.

|  | GExp | LGDest | GSat | GPVal |
|---|---|---|---|---|
| GExp |  |  |  |  |
| LGDest | 0.705[SIG] [0.609; 0.786] |  |  |  |
| GSat | 0.674[SIG] [0.549; 0.766] | 0.882[SIG] [0.801; 0.944] |  |  |
| GPVal | 0773[SIG] [0.681; 0.852] | 0.607[SIG] [0.492; 0.703] | 0.577[SIG] [0.461; 0.682] |  |

Notes: GExp: gastronomic experiences; LGDest: loyalty towards a gastronomic destination; GSat: gastronomic satisfaction; GPVal: gastronomic perceived value.

Consequently, the results at the measurement model level have yielded optimal results, demonstrating validity and reliability at both the indicator and internal consistency or construct levels. The next step is to carry out the analysis at the structural level.

### 4.2. Analysis of the Structural Model

The explanatory power of the model's endogenous variables is determined by the coefficient of determination ($R^2$), which indicates the amount of variance of a given endogenous construct explained by the preceding predictor variables. Authors such as Hair et al. [51] establish weak, moderate, and substantial explanatory levels for $R^2$ values above 0.25, 0.50, and 0.75, respectively. Furthermore, by disaggregating the explanatory power of each of the model's endogenous variables, the explained variance is obtained, which provides a more specific percentage of the variance that each exogenous or predictor variable explains for the endogenous variable it affects.

On the other hand, the effect size ($f^2$) helps explain the degree to which an exogenous construct explains an endogenous construct in terms of the coefficient of determination, so effect size and $R^2$ are directly related. Leading authors such as Cohen [56,57] establish small, moderate, and large effect sizes for $f^2$ values greater than 0.02, 0.15, and 0.35, respectively.

Table 7 shows the results regarding explanatory power and effect size. These results highlight the moderate explanatory power of the two endogenous variables that make up the model, with the greatest explanatory power for the variable "loyalty towards a gastronomic destination" ($R^2 = 0.648$). Regarding the explained variance, gastronomic satisfaction is responsible for explaining 47.72% of the variance of the endogenous variable "loyalty towards a gastronomic destination" and 19.73% of the variance of "gastronomic experiences". On the other hand, it is also worth highlighting the variable "gastronomic perceived value", which contributes to explaining 34.71% of the variable "gastronomic experiences".

The effect size results are consistent with those obtained through explained variance, with the variable "gastronomic satisfaction" having a large and significant power over the variable "loyalty towards a gastronomic destination" and a moderate and significant power

over the endogenous variable "gastronomic experiences". The remaining relationships between variables had no effect.

**Table 7.** Explanatory power and effect size.

|  | $R^2$ | β | Corr. | V.E. | $f^2$(Sig.) |
|---|---|---|---|---|---|
| LGDest | 0.648 |  |  |  |  |
| GSat |  | 0.615 | 0.776 | 47.72% | 0.685(0.000)—Big and significant |
| GExp |  | 0.209 | 0.623 | 13.02% | 0.052(0.115)—Without effect |
| GPVal |  | 0.078 | 0.523 | 4.07% | 0.009(0.473)—Without effect |
| GExp | 0.544 |  |  |  |  |
| GSat |  | 0.335 | 0.589 | 19.73% | 0.187(0.014)—Moderate and significant |
| GPVal |  | 0.512 | 0.678 | 34.71% | 0.435(0.000)—Without effect |

Notes: LGDest: loyalty towards a gastronomic destination; GSat: gastronomic satisfaction; GExp: gastronomic experiences; GPVal: gastronomic perceived value; b: path coefficient; E.V.: explained variance; $f^2$: size effect.

To examine the relationships among the variables, a bootstrapping procedure involving 10,000 subsamples was employed, generating bias-corrected confidence intervals. This non-parametric approach was chosen due to the distributional characteristics of the data (refer to Table 2). Table 8 presents the findings related to the direct and indirect effects, through which the presence of mediating effects is assessed [58,59]. In this context, scholars such as Zhao et al. [60] and Hair et al. [41] suggest that the magnitude of the indirect or mediating effect can be determined using the Variance Accounted For (VAF) metric. Furthermore, Nitzl et al. [61] defines various levels of mediation: no mediation is indicated when the VAF value is below 0.20; partial mediation is present when VAF values fall between 0.21 and 0.80; and full mediation is established when VAF exceeds 0.80 or when the direct effect is nonsignificant while the indirect effect remains significant, representing complete mediation at 100%.

The results obtained reveal the full mediating role of the variable "gastronomic experiences" between perceived gastronomic value" and "loyalty to a gastronomic destination", thus confirming hypothesis 1b (H1b) and, consequently, rejecting hypothesis 1a (H1a). On the other hand, there is no mediating power of gastronomic experiences between the variables "gastronomic satisfaction" and "loyalty towards a gastronomic destination", and, consequently, neither hypothesis 2a nor 2b is supported (H2a and H2b).

**Table 8.** Hypothesis testing. Direct and indirect effects.

| Direct Effects | β | BC CI Bootstrap (95%) | | Results | VAF |
|---|---|---|---|---|---|
|  |  | 2.5% | 97.5% |  |  |
| H1a: C′ (GPVal → LGDest) | 0.078[NSIG] | −0.017 | 0.177 | Not significant |  |
| H2a: C″ (GSat → LGDest) | 0.615[SIG] | 0.511 | 0.705 | Significant |  |
| a1 (GPVal → GExp) | 0.512[SIG] | 0.45 | 0.611 | Significant |  |
| a2 (GSat → GExp) | 0.335[SIG] | 0.224 | 0.454 | Significant |  |
| b1 (GExp → LGDest) | 0.209[SIG] | 0.090 | 0.339 | Significant |  |
| **Indirect Effects** |  |  |  |  |  |
| H1b: a1 × b1 (GPVal → GExp → LGDest) | 0.107[SIG] | 0.045 | 0.189 | Significant | 100%—Full mediation |
| H2b: a2× b1 (GSat → GExp → LGDest) | 0.070[SIG] | 0.031 | 0.123 | Significant | 10.22%—No mediation |

Notes: GExp: gastronomic experiences; LGDest: loyalty towards a gastronomic destination; GSat: gastronomic satisfaction; GPVal: gastronomic perceived value; VAF: Variance Accounted For.

Table 8 shows that gastronomic experiences are essential for transforming perceived value into destination loyalty. This is achieved when visitors experience authentic, sensorially engaging, and meaningful experiences. Mere recognition of the value of a destination's

gastronomy is not enough; it must be accompanied by a direct experience at the destination. Furthermore, the results show a positive relationship between gastronomic satisfaction and loyalty, demonstrating that a tourist who is gastronomically satisfied tends to return or recommend the destination. These results reveal a tourist profile that seeks a more immersive experience, imbued with authenticity at the destination, and, on the other hand, a more functional one.

The final structural model is presented in Figure 2.

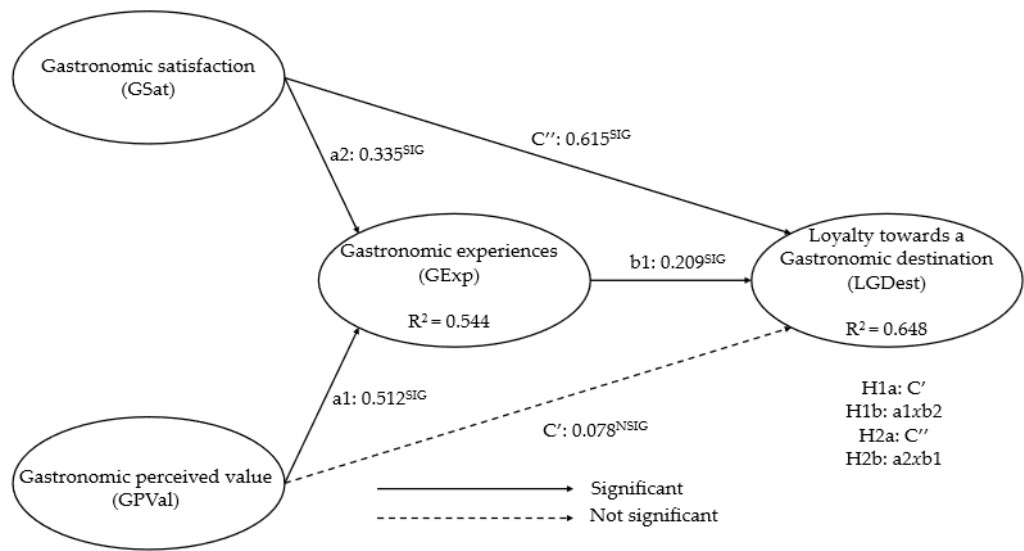

**Figure 2.** Final structural model.

## 5. Discussion

The results obtained in this study confirm and qualify some hypotheses proposed as a result of an exhaustive review of the existing scientific literature. More specifically, the possible convergences are in line with the role of gastronomic experiences as a mediating variable between perceived value and loyalty to a cultural heritage destination such as Córdoba.

This study has demonstrated the mediating power of gastronomic experiences between perceived gastronomic value and destination loyalty. This implies that tourists' perceived value does not translate directly into loyalty, but rather requires an authentic, emotional experience to be positively associated with the tourist and, consequently, with the destination. These results are in line with those previously reported by Taar [17] and Björk & Kauppinen-Räisänen [14], who indicated that gastronomy serves as a means of emotional and symbolic connection with the destination, experienced as a multisensory and differential experience.

Along the same lines, authors such as Jiménez-Beltrán et al. [32] or Ignatov and Smith [31] highlight the component of culinary authenticity as an element that reinforces the relationship between tourist and destination, emphasizing the importance of considering this authenticity from a more experiential point of view that gives rise to future loyalty processes, whether from an attitudinal or behavioural perspective.

Furthermore, the findings of this research show a significant direct effect of gastronomic satisfaction on destination loyalty, confirming H2a. These results are in line with those reported by Hendijani [22], who indicated that greater satisfaction with gastronomy led to a greater willingness to return to the destination and recommend it. Therefore, gastronomic satisfaction is considered a predictor of loyalty, whether recommending the destination or revisiting it. As a result of the above, it can be stated that gastronomic

experiences do not play a mediating role between gastronomic satisfaction and loyalty, and therefore hypothesis H2b cannot be confirmed. In this sense, the findings obtained show that not all tourists consider gastronomy to be a completely integral cultural experience; for many, it is merely a basic need. This implies that this influence is not directly related to the emotional experience lived at the destination [8]. This implies that to achieve destination loyalty through gastronomy, in addition to satisfaction with the food itself, the emotional experiences of the visitor themselves must also be taken into account. Consequently, different motivations, especially hedonistic ones, should be taken into account when determining satisfaction. This, coupled with the fact that in cultural destinations, gastronomy must reflect this cultural component to reinforce the tourist experience.

These results obtained through hypothesis H2b suggest that there is a tourist profile where gastronomy plays a more functional than experiential role, in accordance with the motivations raised by Fields [1] and Quan and Wang [21]. On the other hand, while perceived value requires more emotional components to translate into loyalty, satisfaction acts directly through a more rational experience. This reinforces what was proposed by Rousta and Jamshidi [37], where loyalty is influenced by various experiential dimensions, although not all tourists respond to these stimuli in the same way.

Within the destinations considered World Heritage, gastronomic experiences play a very important role, allowing through them an emotional and symbolic connection between the tourist and the identity of the destination, consuming and understanding the gastronomy of the destination as part of the local cultural heritage, making this experience more meaningful, sensorial, and authentic.

## 6. Practical Implications, Conclusions, Limitations, and Future Lines of Research

### 6.1. Practical Implications

The primary practical implication of this research is to understand how gastronomic experiences and the perceived value of Córdoba's gastronomy determine tourist satisfaction and loyalty. These results allow us to design tourism and culinary products that respond to these tourist demands and, at the same time, are compatible with the sustainable management of local gastronomy. This research aims to enhance the understanding of tourists' perceptions of gastronomy in Córdoba.

### 6.2. Conclusions

Gastronomic tourism is increasingly recognized as a critical strategy for enhancing and consolidating the appeal of various tourist destinations, given the growing interest among travellers in experiencing the culinary traditions of the places they visit. Indeed, for some travellers, the prospect of dining at a particular restaurant or exploring the cuisine of a specific region now constitutes the primary motivation for their journey. Visitors to cultural destinations not only seek to deepen their understanding of the local heritage but also pursue rich sensory experiences. In this regard, gastronomy and its intersection with tourism have emerged as pivotal elements in the analysis of tourist destinations, particularly those grounded in cultural and heritage contexts.

*Conceptually, the findings obtained in this study confirm the need to expand research focused on the role of gastronomic experiences as a driving variable, given that they not only enrich the destination but also modify the way tourists value and process a destination's local gastronomic culture. From an empirical perspective, the results have demonstrated the mediating power of gastronomic experiences between perceived gastronomic value and destination loyalty. This mediating power of gastronomic experiences*

> *cannot be extrapolated to the relationship between satisfaction and loyalty, which is why satisfaction is considered a predictor of behavioural and recommendation intentions.*

*6.3. Limitations and Future Lines of Research*

The limitations of this research include the length of the fieldwork, which would require repeating it over other time periods; the fact that this is a cross-sectional study focused on tourism and gastronomy; and potential biases in the respondents due to the convenience factor. It is also worth noting that another limitation of this research is that the results cannot be generalized to other destinations.

Furthermore, a complete analysis of Córdoba's tourism sector would require parallel research on tourism-related businesses. For this reason, as a future line of research, we recommend conducting in-depth research on the gastronomic supply chain in the city of Córdoba, focusing on tourists. Likewise, future lines of research are proposed, including comparative studies between World Heritage cities with distinct culinary identities, longitudinal studies on destination loyalty behaviour, and research using a methodology that includes ethnographic perspectives.

**Author Contributions:** Conceptualization, C.M.D.-V. and M.d.R.R.-R.; methodology, L.O.-P.; software, L.O.-P.; validation, C.P.-G. and C.M.D.-V.; formal analysis, M.d.R.R.-R.; investigation, C.P.-G.; resources, L.O.-P.; data curation, C.M.D.-V.; writing—original draft preparation, L.O.-P.; writing—review and editing, M.d.R.R.-R.; visualization, C.P.-G.; supervision, L.O.-P. and C.M.D.-V. All authors have read and agreed to the published version of the manuscript.

**Funding:** This research received no external funding.

**Data Availability Statement:** Data are contained within the article.

**Conflicts of Interest:** The authors declare no conflicts of interest.

## Abbreviations

The following abbreviations are used in this manuscript:

UNESCO      United Nations Educational, Scientific and Cultural Organization

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
