# Peer review of "Understanding the Mediating Role of Gastronomic Experiences in a World Heritage Site: An Explanatory Approach to the Case of Córdoba (Spain)"

_heritage, doi:10.3390/heritage8070254_

Round 1

Reviewer 1 Report

Comments and Suggestions for Authors

Thank you very much for giving me the opportunity to review the manuscript entitled “The Mediating Role of Gastronomic Experiences on Perceived Value and Destination Loyalty: A Case Study of Córdoba, Spain.” This article investigates how gastronomy shapes visitor experiences and loyalty in destinations with rich cultural heritage, focusing on Córdoba—the city with more UNESCO World Heritage sites than any other. Drawing on 467 valid questionnaires, the authors employ PLS-SEM to demonstrate that gastronomic experiences mediate the relationship between perceived gastronomic value and tourists’ loyalty. Their findings offer actionable guidance for public and private stakeholders on crafting authentic and novel culinary strategies to strengthen destination attachment.

I especially commend the authors for gathering a substantial on-site sample of actual visitors, which enhances the external validity of their results. The use of structural equation modeling on this robust dataset yields clear evidence of the importance of gastronomic experiences in driving loyalty. However, the abstract would benefit from a more explicit statement of the theoretical framework underpinning the hypothesized mediation, as well as a brief note on the sampling procedure (e.g., timing, survey locations) to clarify representativeness. Additionally, acknowledging the study’s geographic focus in Córdoba—and any implications for generalizing to other heritage destinations—would provide valuable context for readers.

As I read the text of the article, I noticed some aspects that must be improved. These minor revisions are of minor significance. I hope you will find them useful.

These aspects (suggesting minor revisions) are as follows:

  • Literature review: The literature to which the authors refer should be more critical and less narrative. You can show that some specific studies would show better results using the methodologies discussed or any other method / model in your review.
  • Section 3.1 outlines the structure of the questionnaire but does not clarify whether the items—especially those measuring motivations, perceived gastronomic value, satisfaction, and loyalty—derive from previously validated scales or were newly developed for this study. To bolster the rigor and transparency of the instrument, the authors should specify which items were adapted from established, peer-reviewed measures (with appropriate citations) and which were created ad hoc. For any novel items, a description of the pre-testing process (e.g., pilot testing, expert review, cognitive interviews) is needed, along with basic psychometric evidence such as Cronbach’s alpha coefficients or factor loadings to demonstrate reliability and construct validity. The rationale for using polychotomous response formats in the first section versus Likert scales in the second should also be explained, as should the coding strategy for open-ended sociodemographic questions like age. By linking each set of questions back to the literature or documented validation procedures, the authors will significantly strengthen the credibility of their measurement approach.
  • In the Research Methodology section, the inclusion of people under 25 y.o. in the research sample raises questions about ethical principles. Were all participants adults or were minors involved? If minors were involved, were parental consent and supervision ensured during data collection? How were the consent and participation agreements managed, particularly considering the age of the participants? These details should be addressed in the Research Methodology
  • Is It possible to include in the paper the items that were used to measure the perceived value of gastronomy, gastronomic experiences, culinary satisfaction, loyalty to a gastronomic destination.
  • What are the practical or managerial implications of this study?

Author Response

A document with the response to the reviewer is included.

Reviewer 2 Report

Comments and Suggestions for Authors

GENERAL COMMENT

The article “The mediating role of gastronomic experiences in a World Heritage Site. The case of Córdoba (Spain)” addresses a highly relevant and increasingly investigated area at the intersection of heritage tourism and gastronomy.

Focusing on Córdoba, the authors explore the potential of gastronomic experiences to act as a mediating variable in structural relations involving perceived value, satisfaction, and loyalty. The empirical approach—based on a PLS-SEM analysis of 467 valid questionnaires—positions the study within a growing body of literature that seeks to operationalise tourism experiences through quantitative modelling.

Nonetheless, while the study engages with a topic of current academic and policy interest, the manuscript in its present form falls short in several respects. It requires significant revisions in terms of analytical clarity, theoretical grounding, and structural coherence. Notably, the research objectives are not clearly delineated, the literature review lacks critical engagement with recent studies, and the discussion does not sufficiently connect empirical findings to broader conceptual debates in tourism and gastronomy research.

If the authors can effectively address these issues, particularly by sharpening the conceptual framework, justifying the case study more robustly, and deepening the interpretation of results, the manuscript has the potential to contribute meaningfully to discussions on how food-related experiences can enhance the cultural and emotional value of heritage tourism destinations.

Abstract

The abstract requires substantial revision to meet academic expectations. While it introduces the context (gastronomy and heritage tourism) and mentions the methodological approach (PLS-SEM on 467 valid responses), it fails to provide a concise yet informative synthesis of the core findings. For instance, although the mediating effect of gastronomic experiences is mentioned, the abstract does not specify which relationships were supported or how the hypotheses were validated. The abstract should explicitly mention that hypothesis H1b was confirmed (full mediation between perceived gastronomic value and loyalty), while H2b was not. Furthermore, the theoretical and practical implications, particularly for destination managers and policymakers, are missing and should be added. Avoid vague expressions like “can be used by agents” (line 18) and opt for precise contributions grounded in evidence.

Keywords

The keyword section redundantly includes terms already present in the title, such as “World Heritage Site” and “Gastronomic experiences.” It is recommended to replace redundant terms with more specific descriptors that enhance thematic indexing.

Introduction

The introduction outlines the relevance of gastronomy within heritage tourism, highlighting Córdoba’s unique status with four UNESCO recognitions (line 14). However, the section lacks analytical sharpness. It should clearly define:

  • The research gap (e.g., the limited number of empirical studies that test the mediating role of gastronomic experience).
  • The objective of the research is only implicitly mentioned (lines 43–44). A precise statement such as “This study aims to test the mediating role of gastronomic experiences between perceived value/satisfaction and destination loyalty using SEM” should be added.
  • The justification for selecting Córdoba as the case study, supported by socio-touristic and gastronomic indicators, is also missing.

Literature Review

The literature review is structured around four subthemes but lacks analytical depth and does not sufficiently support the hypothesis development. The section “2.4 Gastronomy and Loyalty” is particularly underdeveloped, despite being critical for hypothesis formulation. For example, it does not engage with more recent models of consumer loyalty, nor does it differentiate between affective and behavioural loyalty beyond a definitional note (lines 125–129). This section should be expanded with empirical references from high-ranking international journals.

Section 2.1 (“Gastronomy as a tourist experience”) could be partially moved to the introduction, as it serves more as a general conceptual framing than a review of studies that justify the model tested.

Lines 59–62 (discussion of street food in emerging economies) are not pertinent to the case of Córdoba and dilute the focus of the section. Unless they are used to contrast urban informal food systems with heritage gastronomy, they should be removed.

The review would also benefit from the inclusion of recent studies published after 2020, particularly those using SEM approaches in tourism or studies analysing gastronomy in heritage cities.

Methodology

The methodology is appropriate in general terms but lacks necessary contextualization and transparency.

The authors fail to properly describe the socioeconomic, cultural, and touristic profile of Córdoba, which is essential for understanding the choice of the case and the generalizability of results.

In Section 3.1 (Questionnaire Design), the authors mention the use of Likert scales and polychotomous items, but do not provide the sources or theoretical basis for the items. A table summarizing constructs, number of items, item examples, and literature sources would be advisable. It is also unclear whether the questionnaire was pilot-tested, and how content validity was established.

Section 3.2 (Fieldwork) reports the sample size (n=467) and the period of data collection (June 2022 to June 2023). However, critical information is omitted:

  • Where were the respondents intercepted? (e.g., hotels, heritage sites, restaurants?)
  • Was any form of random or stratified sampling applied?
  • What are the implications of a sample skewed toward young women under 25 (line 192)?

Additionally, under the sociodemographic variables (Table 3), the authors report income brackets, but it is not specified whether these refer to gross or net monthly income, nor the currency standardization for international readers.

Figure 2 (“Relevant sample size”) is embedded as an image, which goes against journal formatting standards. This figure should be redesigned as a vector-based, captioned figure or table.

Results

The statistical analysis is methodologically sound but lacks narrative clarity and interpretive framing. Sections 4.1 and 4.2 alternate between reporting statistical coefficients and brief commentary, but they fail to provide a linear, didactic explanation of the model validation process.

Lines 165–173, which discuss procedural bias and Harman’s single-factor test, would be better placed in a dedicated sub-section titled “Quality Assurance” or “Bias Control Methods”. Presently, they interrupt the flow of the analysis.

The reporting of hypothesis testing in Table 7 is valuable, but the discussion of the results (lines 289–295) is too schematic. While the authors correctly identify a full mediation effect between perceived value and loyalty (H1b confirmed, H1a rejected), and a non-significant mediation effect in the satisfaction-loyalty path (H2b rejected, H2a confirmed), they do not engage with potential reasons or contextual factors for these outcomes.

Furthermore, section 4.2 lacks a visual synthesis of hypothesis testing results, such as a summary table or diagram indicating support vs. non-support.

Discussion

The discussion section (Section 5) reiterates results without sufficiently engaging in a critical comparison with prior research. For example, the confirmation of H1b aligns with studies that highlight the importance of affective engagement in food tourism (e.g., Taar, 2014; Björk & Kauppinen-Räisänen, 2017), but this parallel is only briefly mentioned (lines 311–314) without substantive interpretation.

Crucially, the failure to support H2b (gastronomic satisfaction does not mediate loyalty via experience) is not adequately explained. This is a potentially meaningful finding, suggesting that satisfaction alone may not lead to loyalty unless framed as an emotional experience. A richer interpretation could consider the role of hedonic vs. utilitarian motivations, or cultural differences in the perception of satisfaction.

Overall, the discussion lacks a theorized synthesis of how gastronomic experiences function as cultural mediators, especially in heritage-rich contexts such as Córdoba.

Conclusions, Implications, Limitations and Future Research

This final section is underdeveloped and requires significant expansion. The practical implications are presented vaguely. The authors should explain how destination managers or public authorities could apply the findings—for instance, by designing multi-sensory food experiences, co-branded with heritage sites, to foster emotional engagement and loyalty.

The limitations are superficially addressed. Beyond mentioning the duration of data collection (line 352), the authors should consider; The cross-sectional nature of the study; Potential social desirability biases in self-reported satisfaction and loyalty.; The non-generalizability of the findings beyond Córdoba, particularly due to the sample’s demographic concentration.

Regarding future research, the article merely recommends further investigation into Córdoba’s gastronomic supply chain (line 355), which is disconnected from the conceptual model. It would be more constructive to suggest:

  • Comparative studies between heritage cities with varying gastronomic identities.
  • Longitudinal studies on loyalty behavior.
  • Mixed-method research incorporating ethnographic insights.

References

The reference list is adequate in terms of quantity (49 entries), but not in thematic or temporal coverage. Several cited studies are outdated or too locally focused. There is a lack of recent international literature on food tourism behaviour, loyalty modelling, or the emotional aspects of gastronomy. Inclusion of high-impact articles from journals such as Annals of Tourism Research, Tourism Management, and Journal of Travel Research would enhance academic robustness. Also, key methodological sources on PLS-SEM should be expanded beyond Hair et al. (2014) to include more recent editions and critiques.

Author Response

(The authors gave the same response as above.)

Round 2

Reviewer 2 Report

Comments and Suggestions for Authors

I would like to thank the authors for the substantial revisions carried out following the first review round. The manuscript has improved significantly in terms of clarity, methodological transparency, and theoretical grounding. The integration of updated literature, the clarification of hypotheses, and the improved articulation of the results contribute positively to the overall quality of the paper.

That said, I would recommend three further improvements before final acceptance:

Title or Framing of the Study
Considering the limitations of the research, the authors may consider indicating more explicitly in the title or in the introductory framing of the manuscript that this is an exploratory study.

Discussion Section
The discussion remains relatively brief and, in its current form, does not sufficiently engage with the broader academic debate on gastronomic tourism, cultural mediation, and destination loyalty. I suggest expanding this section to:

  • Better connect the empirical results with prior studies (including some of the recent references already cited in the literature review).
  • Elaborate on the theoretical significance of the rejected mediation hypothesis (H2b), especially in light of distinctions between hedonic vs. utilitarian motivations or between affective and behavioural loyalty.
  • Reflect more deeply on the implications for the conceptualisation of gastronomic experiences as cultural mediators in World Heritage contexts.

Practical Implications Section
Although the conclusion mentions some practical recommendations, I suggest the authors dedicate a distinct and clearly labelled subsection (e.g., “Practical Implications”) before the general conclusions. This would improve the accessibility of the manuscript for destination managers, public decision-makers, and tourism professionals.

Author Response

We thank the reviewer for their comments. We have included a document with a table showing the responses to each observation.
